



# Evaluating dust emission model performance using dichotomous satellite observations of dust emission

Mark Hennen[1]*, Adrian Chappell[1], Nicholas P. Webb[2], Kerstin Schepanski[3], Matt Baddock[4],
Frank D. Eckardt[5], Tarek Kandakji[6], Jeffrey A. Lee[7], Mohamad Nobakht[8], Johanna von Holdt[5].

[1]School of Earth and Environmental Science, Cardiff University, Cardiff, UK.
[2]USDA-ARS Jornada Experimental Range, Las Cruces, NM, 88003, USA.
[3]Institute of Meteorology, Freie Universität Berlin, Germany.
[4]Geography and Environment, Loughborough University, Loughborough, UK.
[5]Department of Environmental and Geographical Science, University of Cape Town, Rondebosch 7701, South Africa.
[6]Centre for Earth Observation, Yale University, USA.
[7]Texas Tech University, Texas, USA
[8]Telespazio UK Ltd, Capability Green, Luton LU1 3LU, Bedfordshire, UK.

*Correspondence to: Mark Hennen (HennenM@cardiff.ac.uk)

## 1. Abstract

Measurements of dust in the atmosphere have long been used to calibrate dust emission models. However, there is growing recognition that atmospheric dust confounds the magnitude and frequency of emission
from dust sources and hides potential weaknesses in dust emission model formulation. In the satellite era, dichotomous (presence=1 or absence=0) observations of dust emission point sources (DPS) provide a valuable inventory of regional dust emission. We used these DPS data to develop an open and transparent framework to routinely evaluate dust emission model (development) performance using coincidence of simulated and observed dust emission (or lack of emission). To illustrate the utility of this framework, we
evaluated the recently developed albedo-based dust emission model (AEM) which included the traditional entrainment threshold ($u_{*ts}$) at the grain scale, fixed over space and static over time, with sediment supply infinite everywhere. For comparison with the dichotomous DPS data, we reduced the AEM simulations to its frequency of occurrence in which soil surface wind friction velocity ($u_{s*}$) exceeds the $u_{*ts}$, $P(u_{s*} > u_{*ts})$. We used a global collation of nine DPS datasets from established studies to describe the
spatio-temporal variation of dust emission frequency. A total of 37,352 unique DPS locations were aggregated into 1,945 1° grid boxes to harmonise data across the studies which identified a total of 59,688 dust emissions. The DPS data alone revealed that dust emission does not usually recur at the same location, are rare (1.8%) even in North Africa and the Middle East, indicative of extreme, large wind speed events. The AEM over-estimated the occurrence of dust emission by between 1 and 2 orders of
magnitude. More diagnostically, the AEM simulations coincided with dichotomous observations ~71%


of the time but simulated dust emission ~27% of the time when no dust emission was observed. Our analysis indicates that $u_{*ts}$ was typically too small, needed to vary over space and time, and at the grain-scale $u_{*ts}$ is incompatible with the $u_{s*}$ scale (MODIS 500 m). During observed dust emission, $u_{s*}$ was too small because wind speeds were too small and / or the wind speed scale (ERA5; 11 km) is

incompatible with the $u_{s*}$ scale. The absence of any limit to sediment supply caused the AEM to simulate dust emission whenever $P\left(u_{s*} > u_{*ts}\right)$, producing many false positives when and where wind speeds were frequently large. Dust emission model scaling needs to be reconciled and new parameterisations are required for $u_{*ts}$ and to restrict sediment supply varying over space and time. Whilst $u_{*ts}$ remains poorly constrained and unrealistic assumptions persist about sediment supply and availability, the DPS data

provide a basis for the calibration of dust emission models for operational use. As dust emission models develop, these DPS data provide a consistent, reproducible, and valid framework for their routine evaluation and potential model optimisation. This work emphasises the growing recognition that dust emission models should not be evaluated against atmospheric dust.

## 2. Introduction

Atmospheric mineral dust has an important impact on many of Earth's systems, human health, and global economies (Li et al., 2018; Pi et al., 2020; Tegen and Schepanski, 2018). The scale of this impact is, at least in part, prescribed by the location and environmental controls of the emission source (Ackerman, 1997; Schepanski et al., 2012). Dust emission models have been developed over decades to resolve spatial patterns and trends of aeolian processes (emission, transport, and deposition) in the dust cycle (Shao et

al., 2011). Dust emission models are also crucial for simulation of aeolian processes at unsampled / monitored locations for comparison with indicators and benchmarks to understand the impact of management on environmental changes (Pi et al., 2020). Dust emission models are also essential for making hindcasts in palaeo-environmental reconstructions (Mahowald et al., 2010) and forecasts in dust-climate interactions in Earth System Models (ESMs).

Global dust emission models were developed more than two decades ago (Marticorena and Bergametti, 1995) and have been rapidly adopted into ESMs, where their fidelity requires necessary compromise and simplification within their parameterisations (Raupach and Lu, 2004). An original constraint, that the Earth's surface was devoid of vegetation and static over time has been partially alleviated with the use of lateral cover but which only very crudely represents the aerodynamics of drag

partition (Raupach, 1992; Raupach et al., 1993). Two key simplifying assumptions remain: i) entrainment threshold at the grain-scale is fixed over space and static over time; ii) sediment supply for transport is infinite and available everywhere. Consequently, the models are well known to over-estimate dust in the atmosphere (Ginoux, 2017). Given the original focus on ESMs, it became common practice to reduce the magnitude of modelled dust emission by comparison with dust in the atmosphere using aerosol optical

depth (AOD; AERONET and dust optical depth). Importantly, AOD is a measure of the concentration of dust in a specific column of atmosphere at a given moment (Dubovik et al., 2000), not a direct measurement of dust emission flux. Additionally, extended atmospheric residence of dust (days to weeks) can exacerbate bias away from dust emission towards atmospheric dust (Schepanski et al., 2012). Consequently, synoptic circulation may increase concentrations within pressure systems, maintaining





AOD over specific areas without any significant further emission (Schepanski et al., 2012). While the deficiencies in existing modelling methods are somewhat understood, the inconsistency of assessing dust emission simulations with atmospheric dust observations prevents a clear direction in how to improve dust emission model fidelity.

The aim here is to demonstrate an alternative to comparing dust emission models to atmospheric dust.
We seek to evaluate the performance of dust emission models against globally observed dust emission at appropriate scales. Our new evaluation is based on two novel approaches. The first is the collation of satellite observed dust emission point source (DPS) data from nine peer-reviewed studies (Baddock et al., 2009; Bullard et al., 2008; Eckardt et al., 2020; Hennen et al., 2019; Kandakji et al., 2020; Lee et al., 2012; Nobakht et al., 2019; Schepanski et al., 2007; von Holdt et al., 2017). DPS data describe
dichotomous (presence=1 or absence=0) dust emission for selected regions and selected times. The second is to determine the coincidence in observed and modelled outcome at each DPS location for every day of the respective study duration, using a contingency table to determine model performance through the respective number of daily Hits (Observed and Modelled dust), Misses (Observed dust, not Modelled), False Positives (Modelled, not Observed dust), and Correct Negatives (no dust Observed or Modelled).
To enable the use of these novel approaches with dust emission models we reduced the continuous dust emission models to the dichotomous occurrence when modelled soil surface wind friction velocity ($u_{s*}$) exceeds the entrainment threshold ($u_{*ts}$). The results of these analyses are compared regionally, with dust emission model performance in different soil-climate environments, demonstrating how modelled and observed dust events coincide over time. These approaches enable us to identify how changes
(incremental or steps) in model development improve model performance related to environmental controls, specifically variability in $P(u_{s*} > u_{*ts})$ and dynamic erodibility of the soil. These analyses provide both i) a robust examination of contrasting dust emission model approaches and, ii) critical information on the fidelity of wind friction velocity thresholds and sediment supply across dust source regions.

We propose this new approach to performance evaluation as the basis for routinely evaluating dust
emission model development and particularly whilst the community is tackling those two key simplifying model assumptions. We recognise that dust emission model developments may not be sufficiently rapid to keep pace with applications e.g., in ESMs, whilst the dust emission models are poorly constrained. Consequently, we also demonstrate how the satellite observed dust emission points (DPS) data are used to calibrate dust emission model estimates to improve their performance. Notably, these calibrations are
against dust emission observations, not atmospheric dust concentrations, providing a more robust description of dust emission model performance.

## 3.  Methods and Data

### 3.1   Validation datasets

We collated nine datasets from established studies across multiple dust emitting regions around the world
(Fig. 1). This global satellite observed dust emission point source (DPS) dataset includes the location and moment of dust emission events from each of the major global dryland areas. For each study, satellite-derived earth observation (EO) data were acquired at regular intervals and subjectively inspected by an



operator to identify the presence of dust plume(s). Identification of elevated dust over a desert surface is particularly challenging in visible wavelengths, due to the spectral similarities of elevated dust and bare

soil in the visible spectrum (Hsu et al., 2004). Therefore, each of the images were converted into false colour composites, enhancing the image with spectral bands outside the visible wavelengths, specifically in the thermal infrared (TIR) bands. Using these dust enhancement products, the operators were able to visually identify the point(s) where a dust plume originated and to digitize each of these locations as a dust emission point source (DPS). The exception is North Africa (Schepanski et al., 2007), where the area

of dust emission is observed sub-daily, within a 1° grid (i.e., frequency of local emission – maximum 1 per day). In this case, the centroid position within the grid box is taken as the dust emission source.

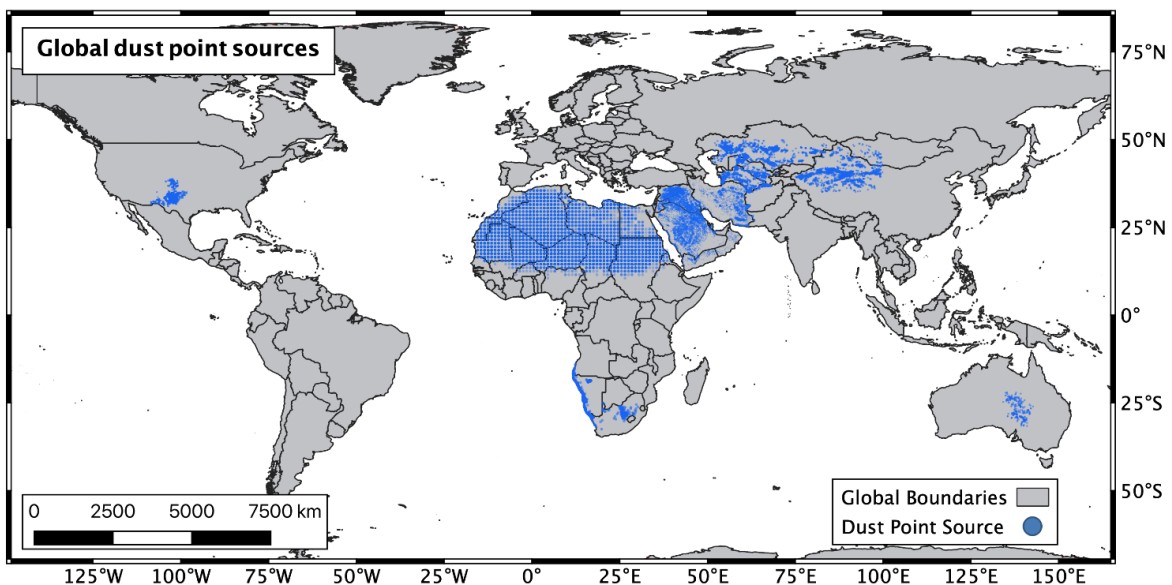

**Figure 1: Global dust emission point sources (DPS), collated from 9 independent studies across 6 dryland environments. Each DPS was subjectively identified in either MODIS or SEVIRI earth observation (EO) data. Data includes a >90,000 individual DPS**
**datapoints, between 2001 – 2016. Source North America: (Baddock et al., 2009; Kandakji et al., 2020; Lee et al., 2012); North Africa: (Schepanski et al., 2007); Middle East: (Hennen et al., 2019); Namibia: (von Holdt et al., 2017), South Africa: (Eckardt et al., 2020), Central Asia: (Nobakht et al., 2021); Australia: (Bullard et al., 2008).**

The methods used during DPS data collection can be classified into two groups, defined by the
type of satellite date used. The majority (7 out of 9) of these studies used Moderate-resolution Imaging spectroradiometer (MODIS) multispectral imagery, which offers twice daily (daylight) imagery of the Earth's surface from two (Aqua and Terra) NASA satellites. These passive optical sensors provide a maximum spatial resolution of 250 m (level 1), recording surface reflectance in 36 individual spectral bands ranging from 0.4 μm (near ultraviolet) to 14.4 μm (TIR) (NASA). Their sun-synchronous orbits
permit repeat observations at the same mean solar time, with Terra and Aqua spacecraft crossing the equator at 10:30 am and 1:30 pm (local time) respectively. For dust plume identification, a dust enhancement product is produced using brightness temperature differences (BTD) between a combination of visible bands (B1 (v. red: 0.645 μm), B3 (v. blue: 0.470 μm), B4 (v. green: 0.555 μm)), near infrared

(NIR) (B26: 1.375 µm) and TIR bands (B31: 11.03 µm and B32: 12.02 µm) to distinguish dust plumes
from the surface and other atmospheric conditions (e.g., clouds, biomass burning) (Nobakht et al., 2019).
These BTDs distinguish the elevated plume as a thermal anomaly from the desert surface below, the
calculated value (dimensionless) is included as the red beam of a RGB false colour composite (FCC)
image, with blue and green beams using visible bands B3 and B4 (Fig. 2a).

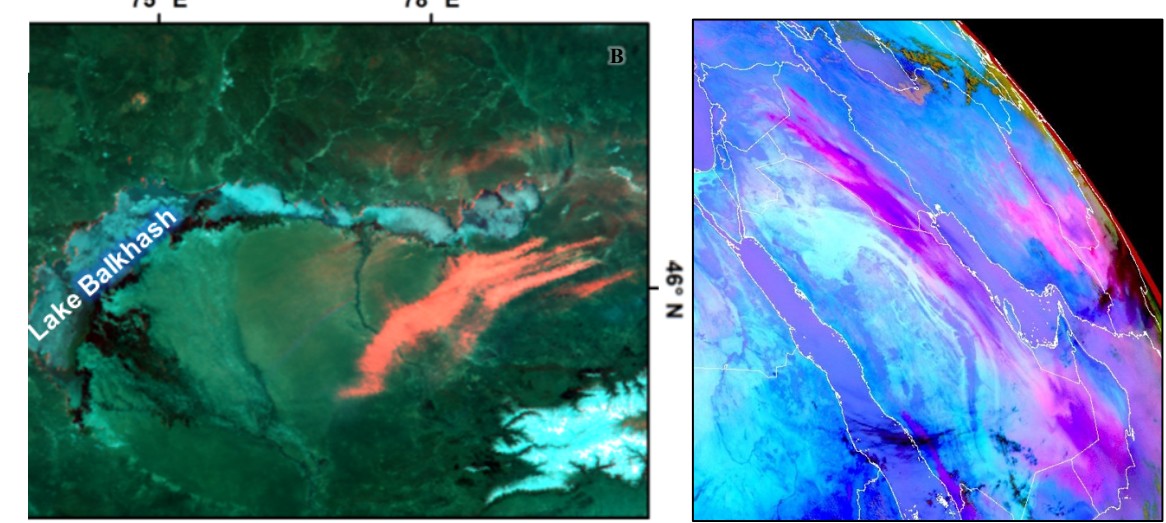

**Figure 2: Dust enhancement products from a) MODIS, RGB bands = R: Dust enhancement (Miller 2003), G: B4 (0.555 µm) and B:**
**B3 (0.470 µm). Source (Nobakht et al., 2019). B) SEVIRI, RGB bands = Red (ΔTBR (12.0µm – 10.8µm), Green (ΔTBG (10.8µm –**
**8.7µm), and Blue B9 (10.8 µm). Source EUMETSAT.**

The two other datasets (North Africa and Middle East) utilise the Spinning Enhanced Visible and Infrared
Imager (SEVIRI) aboard the Meteosat Second Generation (MSG) satellite, which operates in a
geostationary orbit producing frequent (15 minute) repeat observation with a spatial resolution of 3 km at
nadir (EUMETSAT). Like the MODIS DEP, the SEVIRI Dust RGB product identifies atmospheric dust
as a thermal anomaly within the narrow band thermal infrared (TIR) wavelengths (8.7 µm – 12.0 µm)
(Ackerman, 1997; Banks et al., 2019, 2018; Volz, 1973). Compared to clear sky conditions, atmospheric
dust produces a distinctive reduction across each of the TIR channels, with maximum absorption around
10.8 µm (Brindley et al., 2012; Sokolik, 2002). Again, the SEVIRI dust RGB product is rendered through
BTDs, with red and green beams described through the delta between 10.8 µm and adjacent TIR bands
8.7 µm and 12.0 µm, while the blue beam is limited by the BT at 10.8 µm (Lensky and Rosenfeld, 2008).
During dusty conditions, the 10.8 µm channel is suppressed more than the 8.7 µm and 12.0 µm channels,
decreasing BTD (10.8 µm – 8.7 µm), and increasing BTD (12.0 µm – 10.8 µm) and creating a distinctive
pink coloration of dust plumes in the RGB images while clouds appear as red or orange and land surface
as cyan (Fig. 2b).

For each method, absorption in the TIR by water vapour presents a potential limitation, reducing
the cooling trend normally presented by atmospheric dust (Brindley et al., 2012). The presence of
meteorological cloud or dust emission from sources upwind may also prevent observation of the source
of emission in a single image. The 15-minute frequency of SEVIRI data allow the observer to back-track
plume evolution through sequential images to the point of first observation, reducing the impact of





overlapping plumes (Hennen et al., 2019). For MODIS imagery, the 250 m spatial resolution provides finer detail, allowing the observer to better detect individual plume shapes, partially mitigating this overlapping effect (Baddock et al., 2009). Varying surface TIR emissivity occurs due to spatial changes
in surface condition (vegetation, geology), creating variations in the BTD profiles and altering the RGB renderings (Banks et al., 2019, 2018; Banks and Brindley, 2013). During each of these limitation scenarios, subjective interpretation improves upon non-dynamic automated retrieval algorithms, which are required to work in all surface and atmospheric conditions (Schepanski et al., 2012). The shape recognition and decision-making ability of human observation currently exceeds those of automated
approaches, alleviating many of these limitations and preventing false positives observations. For each of these studies criteria are specified for legitimate observation, including i) observation must take place during an emission event, where the deflation surface is clearly identifiable at the head of emission plume; and ii) the distinct dust source must not be obscured by either meteorological clouds or upwind dust emission plumes. As such, these data represent the cutting-edge of dust emission observations, allowing
spatial verification by genuine emission events. The DPS is identified by a presence in dust emission but the absence of dust emission is not recorded (dichotomous). There is an inherent bias in these data towards the occurrence of dust emission and in their quantitative analysis we must account for this bias using statistics designed to handle this bias in dichotomous data.

## 3.2   Dust emission Model

We calculated the albedo-based dust emission (AEM) daily following the established approach (Chappell and Webb, 2016; Hennen et al., 2021). Many dust emission models rely on estimates of saltation flux $Q$ (g m$^{-1}$ s$^{-1}$) to simulate $F$. The $Q$ for a given particle diameter ($d$), soil moisture ($w$), wind speed at height $h$ ($U_h$), and albedo ($\omega$) were calculated as

$$Q(d, w, \omega, U_h) = c \frac{\rho_a}{g} u_{s*}^3(\omega, U_h) \left(1 + \frac{u_{*ts}(d)H(w)}{u_{s*}(\omega, U_h)}\right) \left(1 - \left(\frac{u_{*ts}(d)H(w)}{u_{s*}(\omega, U_h)}\right)^2\right), \qquad \text{(Eq.1)}$$

where $\rho_a$ is air density (1.23 kg m$^{-3}$), $g$ is gravitational acceleration (9.81 m s$^{-2}$), $c$ is a dimensionless fitting parameter (set to 1), $u_{*ts}(d)$ is threshold wind friction velocity (m s$^{-1}$). The soil surface wind friction velocity $u_{s*}$ is the momentum remaining after the removal of momentum by roughness elements at all larger scales (topography, vegetation). The entrainment threshold $u_{*ts}$ (Marticorena and Bergametti, 1995) is described in standard workflows (Darmenova et al., 2009). The $H(w)$ is a function which adjusts
$u_{*ts}$ when soil moisture inhibits entrainment following Fécan *et al.* (1999). The above equation describes how the magnitude of sediment transport is calculated and adjusted by the frequency of occurrence (0 or 1) i.e., $u_{s*} > u_{*ts}$. We used a robust direct estimation (Chappell and Webb, 2016) for $u_{s*}$:

$$\frac{u_{s*}}{U_h} = 0.0311 \left(exp \frac{-\omega_{ns}^{1.131}}{0.016}\right) + 0.007, \qquad \text{(Eq. 2)}$$

where $\omega_{ns}$ is the normalised and rescaled albedo ($\omega$) translated and scaled ($\omega_n$) from a MODIS range
($\omega_{nmin}$=0, $\omega_{nmax}$=35) for a given illumination zenith angle ($\theta$=0°) to that of the calibration data ($a$=0.0001 to $b$=0.1) using the following rescaling equation (Chappell and Webb, 2016):

$$\omega_{ns} = \frac{(a-b)(\omega_n(\theta) - \omega_n(\theta)_{max})}{(\omega_n(\theta)_{min} - \omega_n(\theta)_{max})} + b. \qquad \text{(Eq. 3)}$$



Shadow is the complement of albedo $1 - \omega_{dir}(0°, \lambda)$ and the spectral influences due to e.g., soil moisture, mineralogy and soil organic carbon, were removed by normalizing (Chappell et al., 2018) with the directional reflectance viewed and illuminated at nadir $\rho(0°, \lambda)$:

$$\omega_n = \frac{1 - \omega_{dir}(0°, \lambda)}{\rho(0°, \lambda)} = \frac{1 - \omega_{dir}(0°)}{\rho(0°)}.$$  (Eq. 4)

This was implemented by making use of the available MODIS black sky albedo to estimate $\omega_n$, and the shadow is normalized by dividing it by the MODIS isotropic parameter $f_{iso}$ (MCD43A1 Collection 6, daily at 500 m) to remove the spectral influences:

$$\omega_n(0°) = \frac{1 - \omega_{dir}(0°, \lambda)}{f_{iso}(\lambda)} = \frac{1 - \omega_{dir}(0°)}{f_{iso}}.$$  (Eq. 5)

The $f_{iso}$ is a MODIS parameter that contains information on spectral composition as distinct from structural information (Chappell et al., 2018). In theory, the structural information is waveband independent (Chappell et al., 2018). The normalization of MODIS data using this parameter and that of MODIS Nadir BRDF-Adjusted Reflectance (NBAR) is similarly sufficient to remove the spectral content for all bands examined (Chappell et al., 2018). In practice, we calculated $\omega_n$ using MODIS band 1 (620-670 nm). To retrieve the wind friction velocity as a function of $U_h$, the daily maximum wind speed at $h$=10 metres above soil surface is provided by ECMWF Climate Reanalysis, ERA5-Land hourly wind field data at 11 km spatial resolution (Muñoz Sabater, 2019).

Dust emission $F$ (kg m$^{-2}$ s$^{-1}$) is calculated as:

$$F(d) = A_f A_s M Q(d) 10^{(13.4 \, \%_{clay} - 6.0)}.$$  (Eq. 6)

We allowed $\%_{clay}$ to vary realistically spatially but with the restriction max($\%_{clay}$) =20 common in traditional dust emission models (Woodward, 2001). The proportion of emitted dust in the atmosphere has a relative surface area ($M$) for a given size fraction ($d$). In each pixel, the coverage of snow ($A_s$) and whether the soil surface is frozen ($A_f$) is used to reduce dust emission and is obtained from daily ERA5-Land data. Unlike existing dust models, the use of $\omega_{ns}$ to dynamically estimate $u_{s*}$ removes the need for vegetation indices and fixed vegetation coefficients to determine effective aerodynamic roughness. Furthermore, because $u_{s*}$ is spatially explicit, it is not necessary to apply preferential dust source masks to pre-condition $F$ (i.e., increasing $F$ in areas perceived to have greater erodibility).

Here we tackle the long-standing discrepancy in dust emission model performance being evaluated against dust in the atmosphere (Hennen et al., 2021). Instead, we use satellite observed dust emission point source (DPS) frequency. First, we calculated the DPS probability of occurrence $P$(DPS>0), a first order approximation of the probability of sediment transport P($Q$>0), which is directly proportional to the probability of dust emission $P$($F$>0) at those locations. Next, we equated this to study durations equal to the frequency $u_{s*}$ exceeds $u_{*ts}$ adjusted by $H(w)$:

$$P(DPS > 0) \approx P(F > 0) \propto P(Q > 0) = u_{s*} > u_{*ts} H(w) \begin{cases} 1 \\ 0 \end{cases}.$$  (Eq. 7)

During each simulation, the correct response $(F > 0) \begin{cases} 1 \\ 0 \end{cases}$ depends on the correct $u_{*ts} H(w)$. Importantly, the traditional dust emission schemes, like the AEM used here, assume that the soil surface is smooth and





covered with an infinite supply of loose erodible material which when mobilised by sufficient $u_{s*}$ causes transport and dust emission. This (energy-limited) assumption is rarely justified in dust source regions,
where the soil surface is rough due to soil aggregates, rocks, or gravels, sealed with biogeochemical crusts, or loose sediment is largely unavailable.

## 3.3 Dichotomous testing

At each of the satellite observed dust emission source points (DPS) we used the AEM to predict dust emission daily across the entire time period. The AEM dust emission at these locations were converted
to dust emission occurrence (0 = no dust / 1 = dust) for comparison with the DPS using dichotomous tests. Dichotomous tests are used where the prediction and observation variable contain a maximum of two distinct outcomes. This categorical verification is used in weather forecasting, typically for specific meteorological events (e.g., tornado, rain, or snow), where the verification question is "Did/Will this event occur?" In each instance, observation and simulation will provide a binary response, (i.e., 1 = Yes it
will/did occur, 0 = No it did not / will not occur), these responses can be compared in a contingency table, where the responses are categorised as either Hit (observation=1, simulation=1), Miss (observation=1, simulation=0), False Positive (observation=0, simulation=1) or a Correct Negative (observation=0, simulation=0; **Table 1**). We simulate the presence or absence of dust emission at each DPS location for every day of observation, aggregated at 1° resolution, where if any of the DPS (observed or simulated)
locations produces dust, then that grid box is scored a 1 on that day. Dichotomous statistics compare the coincidence of these 1s. Nan boxes describe lost data due to remote sensing issues (cloud mask, bright pixel mask) are excluded from the analysis. For clarity the number per region are described in the results.

**Table 1: Contingency table describing the frequency of occurrences in the observations and simulations. The joint distribution boxes**
**(Hit, False Positive, Miss, Correct Negative) compare the binary responses of the observations and simulations. The totals describe the marginal distribution for either observation or simulation and are independent of each other.**

|  | **Modelled Yes** | **Modelled No** | **Total** |
|---|---|---|---|
| **Observed Yes** | Hit | Miss | Hit + Miss |
| **Observed No** | False positive | Correct negative | False positive + Correct negative |
| **Total** | Hit + False positive | Miss + Correct negative | Grand total |

We use $P(u_{s*}>u_{*ts})$ to describe the relative conditions of each grid box, with 'windier' locations providing
a higher probability of exceeding threshold. We chose this metric over mean $u_{s*}$ as dust emission is expected to be a rare event (Table 1), obscuring the diversity in extreme wind conditions within the long term mean.





## 4. Results

### 4.1 Satellite observed dust emission point source (DPS) frequency

The frequency of satellite observed dust emission point source (DPS) data and albedo-based dust emission model (AEM) estimates were calculated for DPS locations identified in 6 global dryland regions (using 9 studies). **Table 2** describes the regional results as a probability compared to the number of dust emission opportunities (number of DPS locations multiplied by the number of days in the study minus the number of missing data – see Methods section). Across all 9 studies a total of 37,352 unique DPS locations were
aggregated into 1,945 unique 1° grid boxes, from which a total of 59,688 dust emissions were identified. Missing data ranged from 18.9% (North Africa) to 54.5% (Central Asia), with an average of 34.4% across all nine regions. Corresponding missing data were removed from both modelled and observed data to maintain consistency in results.

**Table 2: Dust event frequency data from dust point source (DPS) locations observed in nine separate earth observation (EO) studies. Data describes the relative probability of occurrence during dust point source (DPS) observation and from albedo-based dust emission model (AEM) forecasts at the same location and time period.**

| | Sensor | Years | Total days (A) | Dust grids (B) | *Missing data (C x 10⁴) | Dust events (D) | DPS P(F>0) (D/(A.B) - C) | AEM P($F$>0) |
|---|---|---|---|---|---|---|---|---|
| **N. Africa** *Schepanski* | SEVIRI | 2006-2010 | 1825 | 927 | 31.9 | 36490 | 0.0266 | 0.18 |
| **Middle East** *Hennen* | SEVIRI | 2006-2013 | 2921 | 431 | 37.5 | 16781 | 0.0190 | 0.42 |
| **Central Asia** *Nobakht* | MODIS | 2003-2012 | 3652 | 398 | 79.2 | 5201 | 0.0079 | 0.22 |
| **Namibian Coast** *vonHoldt* | MODIS | 2005-2015 | 4016 | 36 | 4.9 | 697 | 0.0073 | 0.76 |
| **SW. USA** *Lee* | MODIS | 2001-2009 | 3286 | 13 | 1.7 | 69 | 0.0028 | 0.50 |
| **Australia** *Bullard* | MODIS | 2003-2006 | 1460 | 54 | 1.9 | 148 | 0.0025 | 0.32 |
| **SW. USA** *Baddock* | MODIS | 2001-2009 | 3286 | 12 | 1.3 | 56 | 0.0021 | 0.46 |
| **South Africa** *Eckardt* | MODIS | 2006-2016 | 4017 | 26 | 3.1 | 135 | 0.0018 | 0.12 |
| **SW. USA** *Kandakji* | MODIS | 2001-2016 | 5843 | 48 | 11.5 | 189 | 0.0011 | 0.45 |

*Missing data describes number of simulations (daily grid box) lost due to missing albedo data.

Overall, DPS observations show dust events to be rare, with a regional maximum probability in North Africa of 0.027 or ~10 dust days y$^{-1}$ per 1° grid box (Table 2). In other regions, the probability of dust emission varies, with the Middle East producing the second highest probability (0.019, ~ 7 days y$^{-1}$),



followed by Central Asia (0.008 /∼ 3 days y⁻¹), and the Namibian coast in Southern Africa (0.007, ~3 days y⁻¹). Each of the North American, Australian and South Africa regions produce probabilities >0.003
(~1 day y⁻¹), with the smallest probability of 0.001 (>1 day y⁻¹) in the arid south-west USA (Kandakji). Simulated *P(F>0)* is between 1 and 2 orders of magnitude greater than observations in each region. Furthermore, the relative order between regions varies, with North Africa producing the second lowest probability (P = 0.18 or  ̃65 days y⁻¹), with only South Africa (Eckardt) producing a lower probability (0.12, ∼ 44 days y⁻¹). The Namibian Coast produced the highest probability
(0.76 or ∼ 256 days y⁻¹), followed by North American regions (0.46 – 0.5, ∼ 168 – 183 days y⁻¹), the Middle East (0.42,  ̃153 days y⁻¹), Australia (0.32, ∼ 117 days y⁻¹), and Central Asia (0.22, ∼ 80 days y⁻¹).

## 4.2    Categorical dust emission model performance

The performance of the albedo-based dust emission model (AEM) is assessed through the coincidence of
simulated and observed occurrence (or lack of) dust emission. These results are described globally in **Table 3**, where all results from all regions are collated into a contingency table describing the proportion of each of four outcomes (see Table 1 for outcome descriptions). Dust emission observations account for only 2% of all possibilities (grid boxes multiplied by days). In comparison, the AEM over-predicts the frequency of dust emission by an order of magnitude, producing dust 28% of the time. The model and
observations agree 71.4% of the time, including 0.6% where both model and observations produce dust ('hits'), and 70.8% of the time when neither predicts dust ('correct negatives'). During the remaining 28.6% of the time, the model predicts dust 27.4% of the time when no observations occurred ('false positives') and fails to predict dust 1.2% of the time when observation takes place (**Table 3**).

**Table 3: Categorical statistics for albedo-based dust emission model (AEM) simulations (F > 0) when compared to all satellite observed dust emission point sources (DPS) combined.**

|  | **Modelled Yes** | **Modelled No** | **Total** |
|---|---|---|---|
| **Observed Yes** | 0.6 | 1.2 | **1.8** |
| **Observed No** | 27.4 | 70.8 | **98.2** |
| **Total** | **28** | **72** | **100** |

The variation in modelled dust emission frequency between global regions is explained by the varying cumulative distribution functions (empirical) of wind shear velocity ($u_{s*}$) conditions at the soil surface
(**Fig. 3**). The probability of dust emission (**Table 1**) is defined by the intersection of the distribution of $u_{s*}$ conditions and the entrainment threshold ($u_{*ts}$) of 0.2 m s⁻¹ (in this example; vertical black line), where all simulations greater than that threshold generate dust emission (i.e., *F*>0). In each simulation, $u_{s*}$ is influenced by the soil surface wind friction velocity and surface wind speed ($U_h$). The results show a range of conditions between each of the regions. Along the Namibian coastline (von Holdt) $u_{s*}$ is distinctly
larger than all other regions (mean 0.23 m s⁻¹). In contrast, South African (Eckardt) dust sources have predominantly small $u_{s*}$ (mean 0.11 m s⁻¹) (**Fig. 3a**). In the arid south-west of North America, average $u_{s*}$ remains consistent across each of the three regions (0.19 m s⁻¹), and marginally greater than Australia and





the Middle East (each ~0.17 m s$^{-1}$). Despite producing the same mean, the frequency at which North American regions exceed threshold varies. These regional data suggest that the Chihuahuan Desert
(Baddock), produces a higher proportion of $u_{s*}$ conditions at extreme values (small and large values of $u_{s*}$), whereas the Southern High Plains (Kandakji and Lee) produce a higher frequency closer to the mean. Along with South Africa, $u_{s*}$ conditions in Central Asia (mean = 0.14 m s$^{-1}$) and North African (mean = 0.13 m s$^{-1}$) are the smallest, with $u_{s*}$ values proportionally smaller than the collective global distribution (dashed black line).

**Figure 3b** describe the distribution of $u_{s*}$ conditions during observation periods (locations and days which observed dust only). These data determine the proportion of 'hits' (coinciding observed and simulated dust) by the probability of $u_{s*} > u_{*ts}$. With a greater proportion of $u_{s*}$ values and a $u_{*ts}$ threshold of 0.2 m s$^{-1}$ (vertical black line), the north American regions generate a high probability (0.97-0.99) of 'hits'. In contrast, North Africa, Central Asia, South Africa, and the Namibian Coast all produce 'hit'
probabilities below 0.5, due to the smaller frequency of large $u_{s*}$ conditions. The Middle East (0.55) and Australia (0.84) have higher probabilities but continue to 'miss' a significant proportion of observed dust events. These results show that a high proportion of the observations (up to 79% in North Africa) occur during $u_{s*}$ conditions below threshold, with all regions except North America (Lee and Baddock) producing a minimum observed $u_{s*}$ below threshold.

To demonstrate the impact of $u_{*ts}$ on the probability of dust emission, we consider the adjustment of regional $u_{*ts}$ to match global DPS frequency $P(u_{s*all} > u_{*ts}) = 0.02$, where $u_{s*all}$ is the empirical cumulative distribution functions (ECDF) of $u_{s*}$ conditions at all locations during all days (black dashed line in **Fig. 3a**). Accordingly, adjusted $u_{*ts} = 0.36$ m s$^{-1}$, as the point where the ECDF intersects 98% (blue horizontal line) of the global total distribution (**Fig. 3a**). To assess the impact on model performance and
specifically the ability to simulate dust during observed dust events, this adjusted threshold is applied (dashed vertical line) to $u_{s*}$ conditions during observation periods (**Fig. 3b**). In this case, the percentage 'hits' reduces in all regions, with a maximum reduction of 55% in Australia (84% with $u_{*ts} = 0.2$ m s$^{-1}$, to 29% with $u_{*ts} = 0.36$ m s$^{-1}$), and a minimum reduction of 20% in North Africa (21% to 1%). North America produces the highest percentage of 'hits' (57 – 71%), while all observed events are missed in South Africa
('hit' = 0%). All other regions reduce the proportion of 'hits' below 10%. Overall, during all observed dust events (black dashed ECDF in Fig. 3b), $u_{s*} < u_{*ts}$ 68% of the time, indicating wind speeds are too small over two thirds of the time when we know dust emission has occurred (i.e., DPS > 0).



**Figure 3.** Empirical cumulative distribution functions (ECDF) of satellite observed dust emission point sources (DPS) from 9 studies
across 6 dryland regions compared to MODIS (500 m pixels) albedo-derived wind friction velocity ($u_{s*}$) estimated using ERA5-Land
(11 km pixels) wind speed at 10 m height. The vertical black line represents the model entrainment threshold ($u_{*ts}$) which is fixed
over space and static over time. The distribution of $u_{s*}$ either side of the black line ($u_{*ts}$) represent the probability of modelled dust
emission during a) all modelled days during the duration of the respective study, b) observed days, including only modelled $u_{s*}$
conditions at locations and days where dust point source (DPS) emissions were observed. Red dashed line describes the theoretical
$u_{*ts}$ required to omit 98% (blue horizontal line) of occurrences from the global combined distribution of $u_{s*}$ conditions (black dashed
line), matching the observed frequency of the 9 regional studies (combined) (Table 3).

## 4.3 Dust emission model variability at a local (1°) scale

The ECDF analysis in Figure 3 indicates an underestimation of $u_{s*}$ conditions most of the time during
observed dust events. Accordingly, 68% of known dust events are not modelled. Regionally, this value





varies depending on the range of $u_{s*}$ conditions during observed events (Fig. 3b). Figure 4a describes $P_{obs}$ ($u_{s*} > u_{*ts}$) during observed dust events at a 1° grid box. By simulating only days which are known to produce dust, the results are independent of total frequency, where perfect model performance would produce 1 in each grid box (i.e., every grid box is dark red in Fig. 4a). The variability in grid box $P_{obs}$ is independent of regional conditions (Fig. 3), instead elucidating spatial patterns in $u_{s*}$ conditions during
known dust emission events.

      During observed days, $P_{obs}$ is consistently large (>0.8) across North America, and the southerly reaches of the Lake Eyre Basin, including the Simpson and Strzelecki Deserts to the south (Fig. 4a). Across North Africa, $P_{obs}$ remains generally small, increasing in the north (0.4 – 0.6) along the Mediterranean coast, and decreasing to a minimum (<0.2) in the south and east. In the Middle East, $P_{obs}$
is large (>0.6) across large areas of the Arabian Peninsula, including Mesopotamia in the north, the Red Sea Coast in the west and Oman to the south. Iran has large variability, with $P_{obs}$ (<0.2) in the north-east, increasing $P_{obs}$ (>0.6) in the Sistan Basin to the east, along the Makran Coast to the south and on the shores of the Caspian Sea to the north. Central Asia produces the largest variability, peaking ($P_{obs}$ >0.8) in the Gobi Desert (China) in the east, the Kara-Kum Desert and Aral Sea area (Kazakhstan) to the west,
while many central areas, including the Taklamakan Desert produce small $P_{obs}$. In the Namib Desert, along the Namibian Coast, $P_{obs}$ peaks (>0.6) to the north, while inland $P_{obs}$ reduces significantly (<0.2). In South Africa, $P_{obs}$ is generally small (<0.4), with a peak (>0.6) in the south-eastern extent of the Kalahari Desert.

      For comparison, Figure 4b describes $P_{all}$ ($u_{s*} > u_{*ts}$) during all days at each 1° grid box. These data
include observed events, which only comprise a small proportion (<2%) of all days (Table 3). Accordingly, these results reveal how likely the model is to create false positive dust events, where a good model performance would produce very small $P_{all}$ values (i.e., 0 False positives / each grid box = white Fig. 4b). Here, large spatial variability in $P_{all}$ occurs across Australia, and North America and the Middle East. $P_{all}$ remains consistently small (<0.4) in North Africa, and parts of north-east Iran, Central Asia, and
South Africa. $P_{all}$ peaks (>0.8) in the Namib Desert (Namibia), western Arabian Plateau (Saudi Arabia), Mesopotamia (Iraq / Syria), Makran Coast (Iran), Sistan Basin (Iran/Afghanistan) and discrete parts of the Kara-Kum, Taklamakan (Kazakhstan), and Gobi (China) Deserts.

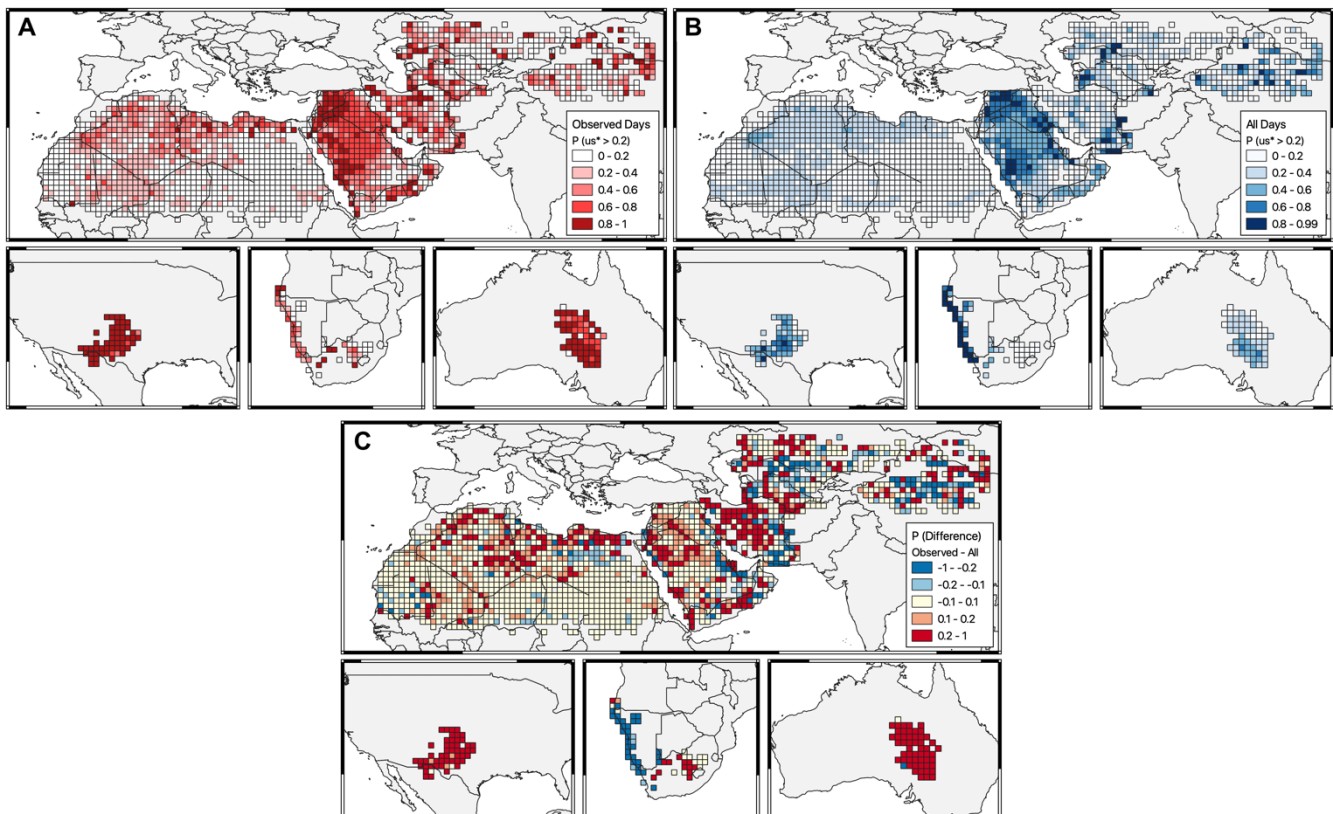

**Figure 4. Maps describing the probability of dust emission ($u_{s*} > u_{*ts}$) at a 1° grid resolution, during (A) observed days and locations where dust point source (DPS) emissions were observed, and (B) all days and locations during the length of the respective study. The difference (C) in $\Delta P$ between observed and all days describes the relative difference in $u_{s*}$ conditions during each period. Red grid boxes describe positive $\Delta P$, meaning winds are larger during DPS dust events than during all modelled days. Blue grid boxes describe negative $\Delta P$, indicating winds are slower during DPS events than during all modelled days. Light blue, yellow, and orange grid boxes described neutral $\Delta P$, indicating none, or very little, discernible difference between wind conditions during DPS events and all modelled days.**

The difference between $P_{obs}$ and $P_{all}$ ($\Delta P$ Eq. 8; **Fig. 4c**) describes how distinct the $u_{s*}$ conditions are in each grid box during each period:

$$\Delta P = P_{obs}\left(u_{s*(observed)} > u_{*ts}\right) - P_{all}\left(u_{s*(a'')} > u_{*ts}\right) \qquad \text{(Eq. 8)}$$

assuming $u_{*ts} = 0.2$. Those $\Delta P$ values close to 0 indicate no, or very small, differences in $u_{s*}$ conditions, indicating that the AEM does not recognise a difference in the probability of $u_{s*}$ exceeding threshold between each period. These conditions occur across large parts of the Sahara Desert, Central Asia, where small $u_{s*}$ conditions continue during all periods ($P_{obs}$ and $P_{all}$ <0.2). In parts of the Arabian Peninsula (including northern Mesopotamia), $\Delta P$ remains small as $u_{s*}$ conditions continue to exceed threshold most of the time ($P_{obs}$ and $P_{all}$>0.6). Positive $\Delta P$ indicates an increase in $u_{s*}$ during observed dust emission days compared to all days. These conditions occur in most dust sources in Australia, North America, Western Arabian Peninsula (Jordan, north-west Saudi Arabia), where $u_{s*}$ conditions are large during dust events ($P_{obs}$ >0.8) and smaller during all days ($P_{all}$ <0.4). $\Delta P$ remains positive in South-eastern Kalahari Desert,



Central Iran, and the Mediterranean Coast of North Africa where smaller $u_{s*}$ conditions during observed dust events ($P_{obs}$ 0.4 - 0.8), remain distinctly larger than on all days ($P_{all}$ <0.2). Negative $\Delta P$ indicates
larger $u_{s*}$ during all days compared to observed days. These conditions occur throughout the Namib Desert, the coast of the Arabian Gulf (Saudi Arabia), the Makran Coast and Dasht-e-Lut Desert (Iran), where $u_{s*}$ conditions exceed threshold most of the time ($P_{all}$ >0.8) and are relatively small during observed dust events ($P_{obs}$ <0.6). Discrete areas of the Kara-Kum, Taklamakan, and Gobi Deserts also produce negative $\Delta P$, as large peaks in $u_{s*}$ conditions ($P_{all}$ >0.8) during all days, exceed those on observed days
($P_{obs}$ <0.6).

## 5.  Discussion

The collective dust emission frequency from nine separate studies demonstrate that dust emission is a rare event (on average 1.8% of all space-time occurrences), even in the more readily recognised dust emission areas (e.g., the Sahara Desert, the Arabian Peninsula), with its infrequent occurrence appearing indicative
of extreme conditions (e.g., high wind speeds). In comparison, AEM simulations estimate dust emission frequency 27% of the time. This over-estimation is expected, as dust emission models are known to be positively biased against dust occurrence (Huneeus et al., 2011). Notably, this over-estimation remains despite the AEM model improvements over traditional approaches using a calibrated attenuation of wind speed by surface roughness (Chappell et al., 2021). There are two components of this AEM dust emission
over-estimation that need to be considered: (i) it is systematic across dryland dust sources around the world; (ii) the disparity is in total frequency and daily coincidence of observed and simulated emission. Without considering both components, it will be difficult for model developments to determine if the model simulated the correct frequency by chance (i.e., same frequency on different days), and under which environmental conditions the model performs.
The AEM coincides with DPS occurrences (observed and not observed) 71.4% of the time. However, during observed dust events, the AEM only coincides with DPS, 32% of the time. Since the AEM provides a realistic (calibrated) representation of $u_{s*}$, these results suggest that the inconsistency in modelled and observed frequencies is due to a combination of three factors: (1) discrepancies in the formulation of the entrainment threshold ($u_{*ts}$); (2) incompatible scales in dust emission modelling,
and (3) the inadequate assumption of infinite supply of loose, fine erodible sediments. Each of these factors can be interpreted by comparing the conditions which exceed entrainment threshold $P(u_{s*}>u_{*ts})$ during observed DPS days to all days (**Table 4**) at multiple scales including, regional (**Fig. 3**) and local (1° - **Fig. 4**).
     The $P(u_{s*}>u_{*ts})$ during DPS events describes the model accuracy in either the $u_{s*}$ conditions known
to have created dust emission (i.e., DPS = 1) or the correct dust entrainment threshold (**Fig. 4a**; top row in **Table 4**). By plotting the combinations of these occurrences, we can understand which meteorological events are best described by dust emission models and / or where the dust entrainment threshold is poorly constrained.  In common with other dust emission models, the AEM has no description of the spatio-temporal variation in soil erodibility and assumes an infinite sediment supply at all locations.
Consequently, whenever $u_{s*}>u_{*ts}$ the AEM simulates dust emission. During DPS observations, by comparing $P(u_{s*}>u_{*ts})$ with all modelled days (**Fig. 4b**), we can determine areas where sediment supply is





poorly described by an infinite sediment supply i.e., no difference in $P(u_{s*}>u_{*ts})$ between observed days and all days (top left and top right in **Table 4**) or comparing $P(u_{s*}>u_{*ts})$ is larger during all days than during DPS observations (bottom left in **Table 4**).

Where there is no clear separation in $u_{s*}$ conditions during observed events and all days, we can interpret these results in two ways, depending on the $P(u_{s*}>u_{*ts})$. If $P$ is large during both periods (bottom right in **Table 4**), the model will correctly simulate dust most of the time during DPS observations ('hits' are large). In this case, dust producing $u_{s*}$ conditions are well described, but the lack of erodibility parametrisation means dust emission will continue to be simulated beyond those days observed in the

DPS data ('false positives' are large). If $P$ is small during both periods (top left in **Table 4**), dust-producing $u_{s*}$ conditions are not well described ('hits' are small) and are therefore not distinguished from all day events during observed DPS days ('false positives' remain small).

Table 4: Description of categorical albedo-based dust emission model (AEM) outputs due to varying probabilities of $u_{s*}>u_{*ts}$ during observed DPS dust days and all days. Colours indicate the symbology applied to 1° grid boxes in Figure 4.

| | | $P(u_{s*}>u_{*ts})$ on observed days (DPS known to occur) | |
| --- | --- | --- | --- |
| | | Small | Large |
| $P(u_{s*}>u_{*ts})$ on all days | Small | **Neutral $\Delta P$**<br>*$u_{s*}$ conditions on observed (DPS) and all days are small and not distinguishable from each other*<br><br>Few 'hits'<br>• Wind field data unable to replicate dust producing winds at DPS locations<br><br>Few 'false positives'<br>• Alterations to $u_{*ts}$ will not differentiate the proportion of Hits and False Positives. | **Positive $\Delta P$**<br>*$u_{s*}$ conditions on observed (DPS) days are distinctly larger than on all days*<br><br>Many 'hits'<br>• Wind field correctly simulates $u_{s*}$ conditions associated with dust emission<br><br>Few 'false positives'<br>• $u_{*ts}$ appropriate for ambient wind conditions. |





| | | Negative $\Delta P$<br>$u_{s*}$ conditions on observed (DPS) days are distinctly smaller than on all days | Neutral $\Delta P$<br>$u_{s*}$ conditions on observed (DPS) and all days are large and not distinguishable from each other |
|---|---|---|---|
| Large | | Few 'hits'<br>• Wind field data unable to replicate dust producing winds at DPS locations<br><br>Many 'false positives'<br>• $u_{*ts}$ inappropriate for ambient wind conditions.<br>• Frequent modelled dust beyond observed days as sediment supply assumed to be infinite. | Many 'hits'<br>• Wind field correctly simulates $u_{s*}$ conditions associated with dust emission<br><br>Many 'false positives'<br>• Alterations to $u_{*ts}$ will not differentiate the proportion of Hits and False Positives.<br>• Modelled dust continues beyond observed days as assumption of infinite sediment supply. |

## 5.1 Discrepancies in the formulation of entrainment threshold ($u_{*ts}$)

By using dichotomous descriptions of dust emission frequency, we provide an assessment of model performance which emphasises the coincidence of events rather than just a comparison of total frequency.
This assessment distinguishes observed simulations from all simulations to provide a powerful description of dust emitting conditions from those on all days. Our results show that modelled dust emission occurs regularly i.e., $u_{s*} > u_{*ts}$ where and when no dust emission is observed (27.4% of all simulations; **Table 3**). These findings suggest that dust emission model performance can be improved by matching $u_{*ts}$ to the correct global frequency of observed dust emissions (globally = 1.8%; $u_{*ts}$ = 0.36 m s⁻
¹). However, reducing the number of 'false positives' in this way will systematically reduce the proportion of correct observations (i.e., 'hits') in all regions by as much as 55% (Australia), with only 1% of all observations in North Africa correctly simulated. An alternative perspective is to adjust $u_{*ts}$ to maximise the number of 'hits' $P(u_{s*observed} > u_{*ts}) = 1$ and globally would require a fixed $u_{*ts} = 0.006$ m s⁻¹. However, this alternative perspective will increase the proportion of 'false positives' to 99.9%.
Despite the rarity of dust observations (occurring only 1.8% of the time; **Table 3**), the ECDF data show that dust emission events rarely represent extreme $u_{s*}$ conditions ($P_{obs} =< P_{all}$; **Fig. 3**), because in most cases there is no distinct difference in $u_{s*}$ conditions between observed days and all days. These results demonstrate that there is no reasonable basis to calibrate model performance through an adjustment to a fixed global $u_{*ts}$. In contrast, a variable threshold should improve model performance in areas where there
is a clear positive change in frequency of occurrence (i.e., top right in **Table 4**; $u_{s*observed} > u_{s*all}$). Our regional results indicate that this condition occurs only in North America, and Australia, where the AEM clearly identifies an increased mean $u_{s*}$ during observed DPS events (**Fig. 3**). In both regions, dust emission occurs during the passage of large frontal systems (Rivera Rivera et al., 2009; Strong et al., 2011), in response to cyclonic activity. The ability to accurately model these synoptic
conditions allows $u_{*ts}$ to be adapted (increased) to reduce the number of 'false positive' simulations



without negatively affecting the model's ability to simulate 'hits'. However, calibration of $u_{*ts}$ in this way is not recommended because it fundamentally tunes the model response to those specific conditions.

## 5.2 Incompatible scales in dust emission modelling

By describing the ECDFs of $u_{s*}$ during observed days and locations (**Fig. 3b**), and assuming that the wind friction velocity normalised by wind speed is well constrained (Chappell and Webb, 2016), a new understanding emerges. Wind speeds used in the AEM are too small to enable $u_{s*}$ to exceed $u_{*ts}$ during roughly 2 out of 3 observed events $P_{obs} (u_{s*} > u_{*ts}) = 0.6$. For example, North American DPS are from predominantly barren conditions and show little variation in $u_{s*}/U_h$, either spatially or temporally (Hennen et al., 2021). This characteristic of DPS data extends globally, with most dust emission source points

coinciding with barren conditions ($u_{s*}/U_h > 0.28$) which do not change much, most of the time (standard deviation less than 0.002) either within or between the few years of measurements. Therefore, variation in $u_{s*}$ conditions of the DPS locations is created mainly by variation in $U_h$. Accordingly, when a dust event is observed but $u_{s*}$ doesn't exceed $u_{*ts}$, we assume that the AEM has not correctly simulated the associated dust-producing wind conditions at that location. In the text which follows, we elaborate on

regional conditions and AEM performance given these assumptions.

Regionally, North Africa produces the smallest probability of dust-producing winds during observed dust events ($P$=0.2). However, **Figure 4a** demonstrates large spatial variability in $P$, with larger values along the Mediterranean Coast and western Africa ($P$>0.4), than inland, eastern parts of the Sahel which have $P$<0.2. Dust emission in the north occurs through cyclogenesis and associated formation of

fronts (Schepanski et al., 2009). Specifically, Sharav cyclones (also named Mediterranean cyclone), track across the Mahgreb region towards the eastern Mediterranean Basin (Knippertz and Todd, 2012). These conditions are often associated with an active warm front, characterised by pronounced dust uplift (Schepanski et al., 2009). Saharan Depressions are also found anticyclonically over Western Africa, where they ultimately transit north and east into a Mediterranean cyclone (Schepanski and Knippertz,

2011). These synoptic scale meteorological conditions are described well in AEM, with a distinct change in $u_{s*}$ (increasing $P$) during observed dust events compared to all days (**Fig. 4c,** top right in **Table 4**).

In parts of the Sahel region, dust emission is associated with mesoscale meteorological drivers, including the diurnal break-down of the nocturnal low-level jet (LLJ) (Schepanski et al., 2009) and sudden increase in wind speeds at the leading edge of cold-pool density currents, formed from deep moist

convection (Knippertz and Todd, 2012; Lawson, 1971). **Figure 4a** shows that neither of these conditions are frequently identified in the AEM, with $P$<0.2 during observed events. These small $P$ values very likely arise from the use of ERA-5 global wind field data (11 km pixels), like most global modelled wind field data, will struggle to describe episodic, mesoscale events such as LLJs and cold pooling (Fan et al., 2020). Instead, these wind data describe a single spatial mean value per 11 km pixel, which is

subsequently used to form $u_{s*}$ and compared to $u_{*ts}$ (at the grain scale without adjustment). The AEM uses maximum daily wind speeds to increase the chance of simulating dust-producing winds. However, maximum values still describe the spatial mean across the 11 km pixel, during that period. If peak wind speeds occur suddenly and/or in a discrete area within each pixel, the mean pixel values will not capture the magnitude of those peak wind conditions at a given point dust source. Accordingly, no distinct change

in peak $u_{s*}$ conditions can be identified during local (discrete) or sudden dust emission events, as





demonstrated by the parity in $P(u_{s*} > u_{*ts})$ during observed dust events and all days (Neutral $\Delta P$ – **Fig. 4c**).

### 5.3 Inadequate assumption of infinite supply of fine sediments.

Despite the AEM failing to simulate all observed dust emission, it over-estimated dust emission frequency
at all dust sources. The dichotomous statistics demonstrate that modelled dust emission occurs predominantly when no observation was made (27% of the time; **Table 3**). At these dust sources, $P(u_{s*} > u_{*ts})$ is large all the time (bottom row in Table 4). As the AEM has no description of the availability of dry, loose fine material to generate sediment transport (soil erodibility), it will produce dust emission whenever $u_{s*}$ conditions are large enough to exceed $u_{*ts}$ (many false positives). The entrainment threshold
is exceeded more frequently in areas where the prevailing wind speeds remain frequently large. Our results show large daily $P(u_{s*} > u_{*ts})$ across Mesopotamia, The Sistan Basin (Iran / Afghanistan) and the Namibian Desert (Fig. 4b), where dust emission is simulated >80% of the time in response to frequent large winds, including the north-westerly Shamal Winds of Mesopotamia (Bou Karam Francis et al., 2017; Yu et al., 2016), the Sistan Winds in eastern Iran (Rezazadeh et al., 2013) and the Berg Winds
across the Namibian coast (von Holdt et al., 2017). The DPS observations peak in some of these regions, yet continue to occur infrequently, with $P(\text{DPS}>0)$ less than 0.3 (see **Appendix 1**). With sufficient wind friction velocity to initiate dust emission 80% of the time, the scarcity of observation indicates an absence of erodible material. Despite an infinite supply of loose material in the model, dryland environments are well-known to be supply-limited (Bullard et al., 2011; Klose et al., 2019; Parajuli et al., 2014; von Holdt
and Eckardt, 2018; Zender, 2003). Ephemeral processes, and the preferential transposition of fine materials are often considered key in the episodic nature of dust emission (Rashki et al., 2017). In supply-limited areas, once these fine materials are deposited, there exists a finite period of increased dust emission potential. During the intervening periods, supply is either exhausted or protected from erosive winds by the formation of biogeophysical crusts (Vos et al., 2020) or surface 'armouring'. Accordingly,
dust source areas, like the Sistan Basin, Tigris-Euphrates Basin (Syria/Iraq), and the Kuiseb River catchment (Namibia), where ephemeral or fluvial systems (with variable flow rates) occur, will tend to be limited by the production of fine materials (von Holdt and Eckardt, 2018). While the impact caused by the simplistic model assumption of infinite sediment supply, is most apparent in frequently windy areas, our results (27% 'false positive' simulations) suggest that the mismatch between the assumption and the
DPS observations of dust emission occurs in all dryland areas (**Fig. 4b**).

### 6. **Conclusions**

Several new insights for model performance have arisen from this work with implications for the prospects of dust emission modelling. Satellite observed dust emission point source (DPS) data, compatible with the scale of dust emission model simulations, demonstrate that dust emission is rare,
even in areas where there are many more dust sources in the region (e.g., North Africa, Middle East). Notwithstanding recent improvements in dust emission modelling using the albedo-based approach, the AEM currently over-estimates dust emission by several orders of magnitude. The over-estimation of dust emission is globally systematic which we interpret here to be due to the consistent difference between the





scale of the wind friction velocity (using MODIS albedo at 500 m) and the scale of wind field data (using
ERA5 Land at 11 km pixels). Similarly, we know that the entrainment threshold is derived at the grain-
scale which is incompatible with those areal estimates of wind and wind friction velocity. Furthermore,
the long-standing dust emission modelling assumption of an infinite supply of dry, loose and available
sediment is evidently unreasonable and causing some of the discrepancy between dust emission modelling
compared with satellite observations of dust emission data. Our results demonstrate that the following
future improvements in dust emission modelling will be most effectively tackled in an integrated approach
because of their interactive nature, by:

- developing an entrainment threshold which varies over space and time, and which is spatially area-weighted to overcome the incompatibility of the current grain (point) scale.

- applying consistently the same spatial scale for all area-based estimates (e.g., wind speed data at 11
km pixels) is now practical by linear scaling of the albedo data to 11 km pixels before it is calibrated to the wind friction velocity and thereby overcoming the non-linearity in sediment transport and dust emission modelling (Raupach and Lu, 2004).

- formulating a parameterisation for sediment supply / availability changing over space which is spatially area-weighted and scales linearly for consistency with other model data.

- establishing values for new model parameters by optimising against satellite observed dust emission (DPS) data.

Model 'tuning' to dust in the atmosphere causes difficulty in routine evaluation of dust emission
model performance. We support the need to ensure that the balance of dust emission modelling is towards
the fidelity of the dust emission scheme (processes) rather than the parsimony of its implementation
(parameterization) (Chappell et al., 2021). As new parameterization schemes are developed and new data
sources become available, the research community will benefit from being open to critical re-evaluations
to avoid model deficiencies enduring. Consequently, rather than making (*ad-hoc*) changes to dust models
by assessment against dust in the atmosphere, dust emission modelling improvements should be made
against satellite observed dust emission point source (DPS) data. True model fidelity will then be
described by the coincidence in space and time with those DPS observations of dust emission, and not
restricted to accumulation/aggregation of frequency.



# 7.  Appendix

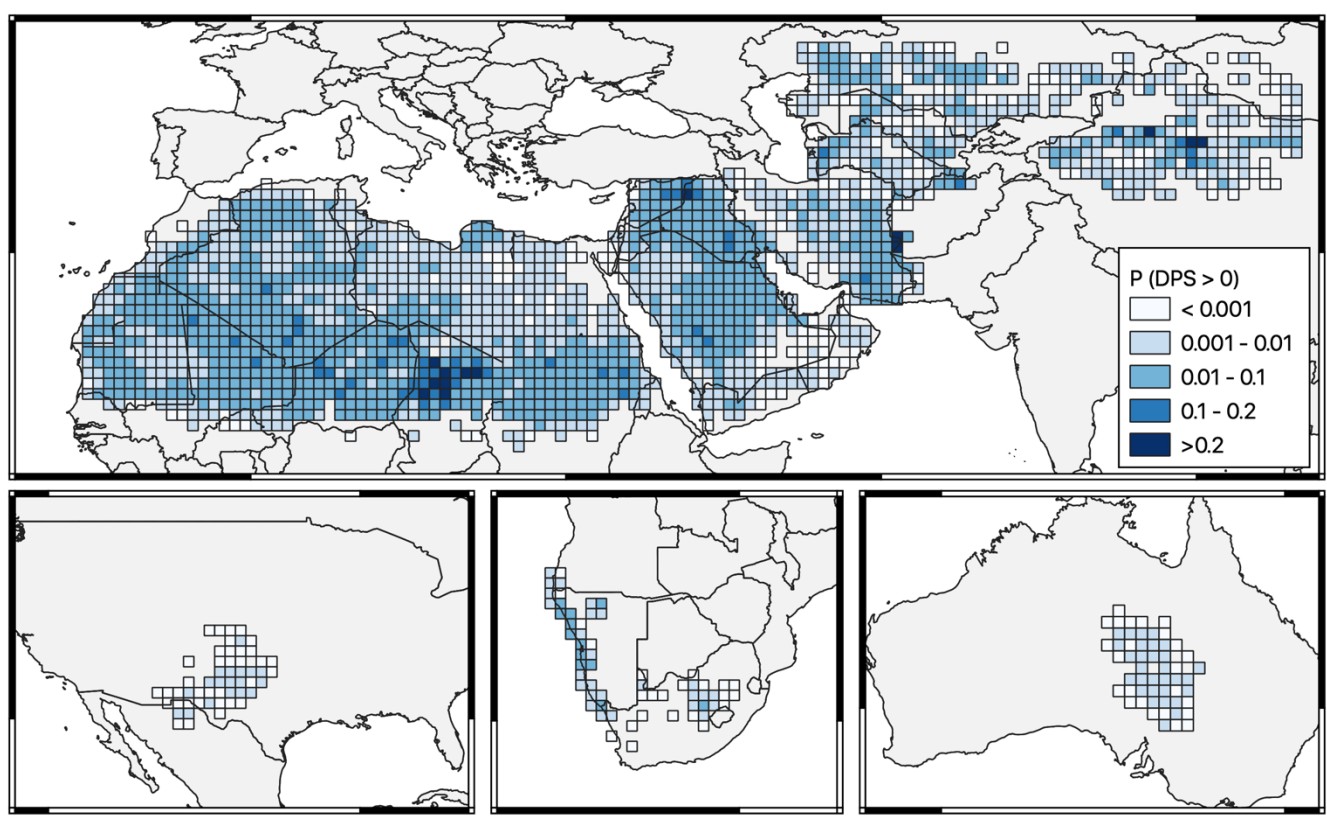

Appendix 1. Probability of dust point source (DPS) observations per day, normalised to 1° grid boxes where frequency is described by a minimum of one DPS observation per day (max. = 0.43). Source North America: (Baddock et al., 2009; Kandakji et al., 2020; Lee et al., 2012); North Africa: (Schepanski et al., 2007); Middle East: (Hennen et al., 2019); Namibia: (von Holdt et al., 2017), South Africa: (Eckardt et al., 2020), Central Asia: (Nobakht et al., 2021); Australia: (Bullard et al.,
605     2008).

# 8.  Code Availability

The Google Earth Engine Java script code and Python code used for the analysis of model output are available through a Zenodo repository (https://doi.org/10.5281/zenodo.5816911).



## 9. Data Availability

The data used are identified in the main text and below using the Google Earth Engine data description and catalogue references, link and DOI. The satellite observed dust emission point source (DPS) data are available from a Zenodo repository (https://doi.org/10.5281/zenodo.5816911).

| Dates used | Google Earth Engine data | Google Earth Engine Catalogue reference, link or DOI |
|---|---|---|
| 2009 | MODIS land cover used to mask land / sea | MODIS/051/MCD12Q1/2009_01_01 https://doi.org/10.5067/MODIS/MCD12Q1.006 |
| Static | ISRIC clay content | https://github.com/ISRICWorldSoil/SoilGrids250m/ |
| 2001-2020 | MODIS albedo Band1_iso | MODIS/006/MCD43A1 https://doi.org/10.5067/MODIS/MCD43A1.006 |
| 2001-2020 | ECMWF ERA5-Land u-component_of_wind_10m v-component_of_wind_10m volumetric_soil_water_layer_1 soil_temperature_level_1 | ECMWF/ERA5_LAND/HOURLY doi:10.24381/cds.e2161bac |
| 2001-2020 | MODIS Snow Cover | MODIS/006/MOD10A1 https://doi.org/10.5067/MODIS/MOD10A1.006 |

## 9. Author Contributions

MH co-wrote the manuscript with AC. MH coded the data analysis and AC and MH performed the analysis jointly. AC coded the dust emission schemes. MH, KS, MB, FE, JVH, TK, JL, and MN all provided DPS data for verification. AC and NPW secured funding from both National Science Foundation and Natural Environmental Research Council (EAR-1853853). All authors contributed to revisions of the manuscript and development of the figures to form the final submission.

## 10. Competing interests

The authors declare that they have no conflict of interest.

## 10. Acknowledgments

The first author is grateful to Google for access to and use of the Google Earth Engine (GEE) and coding support from Noel Gorelick and coding advice from GEE forum members. We thank the following people for useful discussions about the results: Brandon Edwards, Akasha Faist, Gayle Tyree, Brandi Wheeler, and Ronald Treminio. We thank the following organisations for the use of their data: National Centers for Environmental Prediction (NCEP), NASA EOSDIS Land Processes Distributed Active Archive Center (LP DAAC), USGS/Earth Resources Observation and Science (EROS) Center, Sioux Falls, South Dakota; ISRIC SoilGrids; The work was produced whilst MH, AC and NPW were funded by a joint grant from the National Science Foundation and Natural Environmental Research Council (EAR-1853853).





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
