# Peer review of "Evaluating dust emission model performance using dichotomous satellite observations of dust emission"

_Geoscientific Model Development, 2021_

## Author Comment (AC3)

Dear Editor

In our rebuttal (subsequent pages) we show that the reviewer has provided an unbalanced review of our manuscript which makes no mention of our main achievements, including:

- First dust emission model evaluation framework providing the opportunity for all current and subsequent model parameterizations to be evaluated in a concise, open and transparent approach consistent with other disciplines;
- Statistics designed for dichotomous data used to interpret dust emission model skill. The approach places dust emission modelling on to the same rigorous foundation already established for other disciplines e.g., numerical weather forecast model evaluation.

We also demonstrate below that the reviewer has confused our novel approach on dust emission model evaluation with the conventional dust model evaluation against atmospheric dust.

We explain that the reviewer's comments do not recognise that the collation of these DPS data currently provides the most accurate descriptions of dust emission locations and their respective emission frequency. Therefore, these DPS data provide the best opportunity to ground truth the performance of a dust emission model.

We show that the reviewer's description of dust emission point source (DPS) data have "conceptual flaws and limitations" is unjustified and unreasonable. We describe how the validity of the DPS data is long-established by 9 separate publications in high quality, peer-reviewed journals. Collectively those DPS data publications identify all the weaknesses and limitations of the DPS data. In contrast to the reviewer's perspective, we show that our conclusions are unbiased by using these DPS data and that our approach is not limited in its applicability.

The reviewer claims that imperfect DPS data is the main cause of bias which is misleading our conclusions. However, we explain in our manuscript and here that dust emission models are strongly related to DPS data. The bias is caused by the well-known tendency of dust emission models to over-estimate dust emission caused by unrealistic dust emission model assumptions about entrainment and infinite supply of sediment everywhere.

Our rebuttal demonstrates that our results are valid and that our new approach stands as a valuable opportunity for the community to robustly evaluate dust emission models. We accept that there are some parts of our text which can be clarified. For example, we believe that re-wording parts of the conclusion will provide additional clarity on our stated position, and any specific concerns with the use of DPS data can be addressed in the methodology.

Regards,

Dr. Mark Hennen (on behalf of the authors)

Dear Reviewer,

We appreciate you taking the time to provide comments on our manuscript. We provide below, responses to your comments using this purple coloured text. We start with the following paragraph since your first paragraph was largely a repeat of our abstract.

While the overall approach could represent a valuable complement to current evaluation capabilities of dust models, its current implementation and interpretation has, in my opinion, several conceptual flaws and limitations that very likely bias the conclusions of the paper and largely limit the applicability of the approach. In addition, in view of these limitations, statements such as "the study emphasizes the growing recognition that dust emission models should not be evaluated against atmospheric dust" are just not supported by evidence provided in the paper.

To conflate dust emission with atmospheric dust is to hide the processes which are poorly constrained in the dust emission modelling. We think this issue is well illustrated in Figure A1. The green dots are dust emission source points (DPS) data from which the downstream dust (brown area) then influences the dust optical depth. The dust will not stay close to source but will remain in the atmosphere depending on the particle size of the emitted dust and atmospheric turbulence and will only disperse with large wind speeds. Whilst the dust remains aloft, more dust emission can occur from other sources which would readily accumulate atmospheric dust. In this simple description you have the parsimonious basis for how North Africa is so very dusty but with so few dust days per unit area.

[Figure]

Figure A1: An observed example of dust point source (DPS) identification from MODIS true colour imagery in northern Texas on February 24 2007. Source (Kandakji et al., 2020).

Figure A2 is taken from Hennen et al. (2022; Figure 3) and demonstrates the relation for North American dust emission point source (DPS) data and dust emission model data.

[Figure]

Figure A2. Comparison of observed and modelled dust emission frequency ($F > 0$) and magnitude ($F$). The albedo-based annual (2001–2020) average dust emission frequency (A) at dust point source (DPS) locations (i.e., panel B) on observed DPS days (y-axis) and all days during observation period (x-axis). The dashed line is the 1:1 line. The inset plot shows the validation of the calibration function fitted to albedo-based dust emission values. Uncalibrated modelled average annual dust emission frequency (B) in the region of the DPS locations. DPS locations include results from Lee et al. (2012); Baddock et al., 2011, Kandakji et al., 2020.

Figure A3 below is from our complementary manuscript (under review in Environmental Modelling & Software) which demonstrates for a North American region how dust emission point source (DPS) data are poorly related to dust optical depth (DOD; orange dots and line).

[Figure]

Figure A3. Modelled and observed frequency at known North American satellite observed dust emission point sources (DPS), identified in satellite observations (Kandakji et al., 2020; Lee et al., 2012; Baddock et al., 2011). For each point, the y-axis represents the observed number of DPS observations (per grid cell) per year during different observation phases of the DPS datasets within the observation time period (2001 – 2016). For AEM and TEM, the x axis describes number of modelled observations (F>0) at DPS locations in each grid cell per year during the same time period (x-axis). For DOD, the x-axis describes the frequency that DOD>0.2 per year for the same period. The least squares logarithm regression of modelled against DPS observations produced the model parameter coefficients, $R^2$ correlation and the square root of the mean squared difference between DPS, and model predictions (RMSE) adjusted by the degrees of freedom using the number of dust emission model parameters (df = 9 for AEM; 12 for TEM; 6 for DOD).

Below are my general comments/concerns. Based on them I do not recommend publication of this manuscript as I cannot see how those fundamental limitations and their impact on the conclusions and perspectives of the paper can be easily amended.

As we mentioned at the outset of our responses, our concern with the reviewer's comments is that they do not represent what the manuscript has achieved. The reviewer's comments focus solely on perceived weaknesses of dust emission point source (DPS) data. As we have shown in the response to the previous point, these DPS data are well-established in the literature. Therefore, we do not find that the reviewer's comments are representative of the community. Consequently, we do not accept that the reviewer's comments are a valid basis to reject our manuscript.

**On the evaluation of models with observations of dust emission point sources vs atmospheric dust:**

Our manuscript does not evaluate "models" (unqualified) as stated above, our manuscript evaluates a dust emission model, and our results suggest that dust emission models should not be evaluated against atmospheric dust.

Satellites do not observe dust emission but atmospheric dust. Estimates of dust emission point sources are retrieved or inferred from atmospheric observations. In fact, the same applies to in situ measurements: emission cannot be observed directly and can only be inferred from airborne measurements. This is an important conceptual nuance, and it must be clear that the proposed framework relies on a DPS dataset, which infers emission point sources based on many assumptions and potentially important limitations as I will describe below.

This comment suggests a misunderstanding on the formation of these dust emission point source (DPS) datasets. We agree that satellites observe atmospheric dust. The important nuance here, is that by expert inspection of satellite imagery, it is possible to identify dust plumes and trace over space-time the plumes to the location from which they were emitted (Figures A1, A2 & A3 above).

Dust emission typically occurs infrequently (as described in the published DPS data; e.g., Hennen et al., 2019), and in remote and inhospitable areas. Field measurements of their occurrence rely either on a limited number of ground stations or serendipitous observations. For these reasons, satellite-based remote sensing is ideally positioned to monitor emission and identify the source of these emissions.

Currently, automated approaches are not well-established to accurately distinguish the point at the head of the plume (i.e., the dust emission point source). Therefore, DPS identification is performed by expert analysis, where an expert observer can study the shape of the plume, recognise any atmospheric opacity (clouds, smoke, dust, or fog) and pinpoint the location of the source.

These DPS data are all obtained from previously published, frequently cited, peer-reviewed journals, which are accepted in the community as an accurate description of dust emission activity. Therefore, these data currently represent the most robust set of dust emission observations from which to evaluate the performance of a dust emission model.

Figure A1 (above) is an example from that DPS literature and shows the dust emission point sources (DPS; green dots). At the time these data were produced, the location of those dust emission sources was identified by individual inspection of images.

Hennen et al., 2019 An assessment of SEVIRI imagery at various temporal resolutions and the effect on accurate dust emission mapping. *Remote Sensing*., 11 (8) (2019), p. 918, 10.3390/rs11080918

Even if the implementation of the proposed evaluation approach would be sound, why dust emission models "should not be evaluated against atmospheric dust"? Why is it incompatible?

Dust emission modelling is a key initial phase of dust (cycle) models which include all phases of the dust cycle. The dust emission models describe the momentum transfer through roughness canopies to establish the remaining momentum which can exceed the sediment entrainment threshold. At the point of emission, the dust emission model is complete. To evaluate how well that dust emission model has described the dust emission process requires knowledge of whether the dust emission has occurred or not, which is evident in the dust emission point source (DPS) data. In contrast, to evaluate the dust emission model against dust in the atmosphere is to inappropriately expect *a priori* that the factors controlling the dust cycle (dust emission, transport, residence and deposition) are useful descriptions of dust emission (which they are obviously not). Hence, we are clear and correct in our approach: dust emission models should not be evaluated against atmospheric dust.

The statement is just not justified, particularly given the limitations highlighted in my next comment. Evaluation efforts of dust models (with embedded emission and dust cycle) typically include a variety of observations (in situ, satellite, remote sensing) of different

variables at different spatial and temporal scales, and all are very welcome and helpful to characterize the behavior of a dust model including its emission. Why not seeing different approaches as complementary? In any case, the statement is just an opinion and is basically not supported by the results of the paper.

We do not challenge how dust (cycle) models are evaluated. We recognize the benefits of calibrating dust models to atmospheric dust. The reviewer appears to have confused our focus on dust emission model evaluation with that of dust (cycle) modelling.

We are clear in our statements in the manuscript and here: dust emission models should not be evaluated against atmospheric dust, particularly when dust emission models are readily evaluated against dust emission point source (DPS) data (Hennen et al., 2022).

Unless or until dust emission models are evaluated correctly against dust emission, we will not be able to establish whether essential dust emission parameterisations are improving performance in dust (cycle) models. This is the essential purpose of our manuscript.

**On the use of DPS datasets to evaluate dust emission models:**

At present there are well known limitations in the retrievals used to infer dust emission that are very likely strongly biasing the comparison with the dust emission model.

We have already published (Hennen et al., 2022) evidence that dust emission models show strong relations with DPS data, but those dust emission model estimates are several orders of magnitude too large (see Figure A2 above).

We have also provided evidence above that dust emission models are unrelated to dust in the atmosphere (measured by dust optical depth; Figure A3 above). In our previous response above, we have explained how that mismatch is intuitively reasonable.

Recall that dust emission models are expected to over-estimate dust emission. As we described in our manuscript, the limitations of dust emission models are that they assume unrealistically that the sediment entrainment threshold is fixed over space and static over time and that loose erodible sediment is supplied infinitely across every ground surface. These assumptions are causing the strong bias and the orders of magnitude over-estimation in modelled dust emission at far too many locations. This is entirely consistent with what the DPS data are showing, and we have already published (Hennen et al., 2022).

Take for example SEVIRI Dust RGB product used over North Africa and the Middle East. It is well known that the product can detect the particularly high-concentration dust storms but fails in detecting thin, low level and/or low-medium concentration dust clouds/events,

which can come from frequent low emission events that are widespread in North Africa and the Middle East (partly due to the high availability of saltators).

We do not pretend that the dust emission point source (DPS) data are perfect. We expect there to be errors, many of which we have described in our manuscript in an open and transparent approach.

There are two key points which are being missed by the reviewer's focus on DPS data:

1. We expect that the evaluation of the dust emission model against (imperfect) DPS data to be better than evaluating the dust emission model against atmospheric dust for all the reasons we described above (evident in Figure A3).
2. The same uncertainties of DPS data are common in aerosol / dust optical depth. However, there is widespread acceptance in the literature and in the reviewer's own comments that dust (cycle) models are evaluated against these (imperfect) data. It seems contradictory to claim that the imperfect nature of DOD data is acceptable for dust (cycle) model comparison, but that the imperfect nature of DPS data is unacceptable and produces bias when compared with dust emission models.

This important limitation invalidates to a large extent the proposed dichotomous evaluation approach as dust emission from the model contains all type of dust emission events (us* > u*ts) and dust emission from the DPS is strongly biased towards high emission events, which makes the proposed framework currently inconsistent, and the conclusions likely flawed.

We agree that some satellite sensors may not identify the dust emission point sources (DPS) of all types of dust emission. We note that this same criticism is true of aerosol optical depth (AOD) and dust optical depth (DOD). However, there is a long history of publications which evaluate dust (cycle) models against these AOD and DOD data. Consequently, we find no substance to the reviewer's claim made here that our conclusions are likely flawed.

For example, the overestimation of the occurrence of dust emission in the AEM by 1 – 2 orders of magnitude and the rarity of dust emission in the DPS for North Africa and the Middle East point towards a problem in this sense.

As we explained in the manuscript, the difference of 1-2 orders of magnitude difference in the frequency of modelled dust emission relative to the dust emission point source (DPS) data is caused to the first order by the crude assumptions of sediment entrainment and an infinite supply of sediment everywhere. Our findings are explained and intuitively reasonable when considering the dust emission model alone.

Another potential problem for the evaluation of global models is the inconsistency of the DPS among regions. Using MODIS in some regions and SEVIRI in others with their different sensitivities could further bias the conclusions on the behavior of a model in different regions.

We make no comment about the evaluation of global "models" (unqualified). Our approach is for dust emission models.

In contrast, our manuscript describes, and we explain here, how we explicitly tackle the requirement to harmonise the DPS data over space-time by aggregating using grid boxes to the largest SEVIRI resolution.

Any issues that "could further bias" the conclusions based on difference are already accounted for by transforming the reflectance data using a threshold which produces a dichotomous response. The same transformed approach is used with the dust emission model estimates. This approach ensures consistency within and between grid boxes across regions.

The reviewer claims that differences between sensors used in different regions differently influence our results. However, there is no basis provided for this claim, it is unsubstantiated.

It is very easy for the reviewer to suggest that some uncertainty "could" bias the conclusions. Unless there is either a strong logical basis or some quantitative evidence, the reviewer's comment remains a hypothesis to be tested, not an assessment of our manuscript.

A nice exercise would be to compare a DPS in North Africa and the Middle East based on SEVIRI with another one based on MODIS, and see how the evaluation and the conclusions are impacted.

This could be an interesting exercise. However, it is effectively examining the influence of resolution on the outcomes. As we explained in the previous response the influence of resolution / commission difference is removed by aggregating the DPS data in to common SEVIRI-sized grid boxes. In any case, we think that our manuscript is valid and describes an evaluation framework which would enable the community to debate any arising issues.

There are no easy solutions to circumvent the problem of the low-medium dust emission events. Also, this problem clearly evidences that quantitative AOD products (along with their quantified uncertainties) over sources regions can and should at least complement dust emission model evaluation efforts.

Agreed, and which is why we found it unreasonable when earlier you used DPS errors as the basis for the criticism of our work. Our work is attempting to tackle the much larger issue that dust emission modelling has very crude, unrealistic assumptions about sediment entrainment and infinite sediment supply. These assumptions have existed for more than two decades since dust emission models were first developed. These weaknesses endure because the dust (cycle) models circumvent these dust emission (and other component) limitations by tuning modelled dust to atmospheric dust.

Here, in the manuscript and our recent publications (Hennen et al., 2022), we have explained that it is evidently and intuitively reasonable that modelled dust emission does not match atmospheric dust. Whilst dust emission modelling remains poorly constrained and unrealistic, it is very unlikely to represent the correct dust magnitude and frequency. Under these current conditions, tuning the dust (cycle) model to dust in the atmosphere simply adopts the (more or less) incorrect dust emission.

Here, in our manuscript, and our publications (Hennen et al., 2022) we are interested in ensuring that the dust emission modelling is correct. When the corrected dust emission is then used in a dust (cycle) model it will inherit the correct magnitude and frequency before being tuned against atmospheric dust. This improvement is expected to make more reliable historical or future projections.

In addition to these inherent biases in the DPS and the associated comparison, the difference between simulated wind scale, and the DPS scale makes the interpretation of the results very complex. I acknowledge there is a section in the discussion about this problem but in my opinion the issue should already be considered in the basic design of the evaluation framework. In other words, models should be evaluated as much as possible at consistent spatiotemporal scales, otherwise conclusions can be fundamentally flawed.

For clarity, we accept that the DPS data are not perfect descriptions of dust emission and that they have similar uncertainties to AOD and DOD data. We have shown above that DPS data are strongly related to dust emission model estimates and 1-2 orders of magnitude over-estimated frequency is caused by dust emission model assumptions that entrainment threshold is fixed and static over time and that there is an infinite supply of sediment everywhere.

We state clearly (Section 3.3 Dichotomous Testing) how the dust emission model is aggregated and how the DPS data are aggregated e.g.,

"We simulate the presence or absence of dust emission at each DPS location for every day of observation, aggregated at 1° resolution, where if any of the DPS (observed or simulated) locations produces dust, then that grid box is scored a 1 on that day."

Consequently, the issue of scale is already considered in the basic design of the evaluation framework. We did this precisely to ensure consistency in the spatio-temporal scale of these difference data. On this basis we do not accept that our results are flawed.

We believe the reviewer has misunderstood the text in the Discussion (Section 5.2 L. 495-532). It is not about the methodology used in the comparison of the data. As stated in the title of that section, the text is about "Incompatible scales in dust emission modelling".

All in all, the previous highlighted limitations are very likely biasing the results obtained and the derived conclusions. For example, the overestimation of the frequency of dust emission is partly attributed to the absence of any limit to sediment supply.

The reviewer has provided some opinions about dust emission point source (DPS) data and based on those opinions has suggested our results are very likely biased. Whilst we appreciate the opinions, we have shown that those opinions are not representative of the wider community understanding and value of those DPS data. More fundamentally, we have shown that those opinions include the achievements in our manuscript. Consequently, those opinions about DPS data do not represent a review of our manuscript. Therefore, we do not accept that these opinions provide a valid basis to evaluate our manuscript.

Regards,

Dr. Mark Hennen on behalf of all authors